Online teaching emotion analysis based on GRU and nonlinear transformer algorithm

Ding Lan 2017360002@xyafu.edu.cn
College of Tea Science, Xinyang Agriculture and Forestry University , Xinyang , China
Alatas Bilal
Electronic publication date: 2023 Nov 21
Publication date: 2023
Volume: 9
Electronic Location ID: e1696
Received 2023 Sep 15; Accepted 2023 Oct 23
Copyright: © 2023 Ding
Copyright year: 2023
Copyright holder: Ding
License: This is an open access article distributed under the terms of the Creative Commons Attribution License, which permits unrestricted use, distribution, reproduction and adaptation in any medium and for any purpose provided that it is properly attributed. For attribution, the original author(s), title, publication source (PeerJ Computer Science) and either DOI or URL of the article must be cited.
License URL: https://creativecommons.org/licenses/by/4.0/

Keywords: Sentiment analysis, Transformer, Online teaching, GRU

Funding: The author received no funding for this work.

==============================
Nonlinear models of neural networks demonstrate the ability to autonomously extract significant attributes from a given target, thus facilitating automatic analysis of classroom emotions. This article introduces an online auxiliary tool for analyzing emotional states in virtual classrooms using the nonlinear vision algorithm Transformer. This research uses multimodal fusion, students’ auditory input, facial expression and text data as the foundational elements of sentiment analysis. In addition, a modal feature extractor has been developed to extract multimodal emotions using convolutional and gated cycle unit (GRU) architectures. In addition, inspired by the Transformer algorithm, a cross-modal Transformer algorithm is proposed to enhance the processing of multimodal information. The experiments demonstrate that the training performance of the proposed model surpasses that of similar methods, with its recall, precision, accuracy, and F1 values achieving 0.8587, 0.8365, 0.8890, and 0.8754, respectively, which is superior accuracy in capturing students’ emotional states, thus having important implications in assessing students’ engagement in educational courses.

Introduction

With the swift evolution of the Internet, online classrooms have emerged as a potent pedagogical approach, supplementing traditional instructional methods (Salas-Pilco, Yang & Zhang, 2022). Particularly during the years marked by the epidemic, a substantial proliferation of online classroom education has superseded conventional classroom education, playing a noteworthy role in society’s triumph over the outbreak (Nambiar, 2020). Emotional analysis holds paramount significance in traditional and online classroom education, as it gives insights into students’ emotional states, enabling a timely understanding of their knowledge acquisition and learning dispositions. Consequently, this analytical process enhances classroom efficacy and teaching efficiency (Li et al., 2023). However, discerning students’ emotions in the online classroom presents a more significant challenge than in traditional teaching, as the absence of a shared empathetic environment for teachers and students hinders such endeavors (Tang et al., 2023). Investigating how to accurately analyze and regulate students’ emotions within an online classroom constitutes a topic of significant research importance.

Sentiment analysis encompasses the processes, techniques, and methodologies employed to automatically ascertain individuals’ attitudes or emotional orientations expressing their opinions on a given subject matter through textual discourse (Chan et al., 2023). Conventional sentiment analysis approaches focus on textual modality, where word meanings and logical connections within statements are established to decipher sentiment (Birjali, Kasri & Beni-Hssane, 2021). However, unimodal sentiment analysis methods are susceptible to dichotomous interpretation, frequently resulting in misjudgments of sentiment expressions across diverse contexts. Propelled by the progressive substitution of unimodal text data with multimodal data on the Internet in recent years, coupled with the advantage of synthesizing the sentiment polarity of identified objects across various dimensions, multimodal sentiment analysis (Kaur & Kautish, 2019) has emerged as a burgeoning area of research interest.

Multimodal sentiment analysis methods can be classified into two categories: interactive multimodal sentiment analysis methods and narrative multimodal sentiment analysis methods (Meng et al., 2022). The former focuses on capturing conversational thought shifts and content uncertainty, necessitating rigorous training data requirements. Conversely, the latter concentrates on decision-based sentiment analysis in identified objects using fixed data. Narrative multimodal sentiment analysis is further subcategorized into decision and feature-level (end-to-end combinatorial) analyses. Decision-level analysis (Zhang et al., 2022; Yang, Na & Yu, 2022; Yang, Li & Zhang, 2021) can be implemented using various combinations of existing models such as Transformer, convolutional neural network (CNN), long short-term memory (LSTM), etc. These models offer the advantages of simplicity and flexibility, yet training their parameters demands additional resources and time investments. End-to-end combinatorial analysis models leverage existing models and optimization algorithms to address training-related challenges.

For instance, in Xiao et al. (2020), a hierarchical attention module inspired by Transformer’s multiheaded attention mechanism resolves attention attenuation concerns. In Huddar, Sannakki & Rajpurohit (2021), an emotion detection module inspired by the lightweight attention aggregation module tackles the problem of emotion information loss during training. Additionally, Huddar, Sannakki & Rajpurohit (2020) proposes a modal internal fusion module referencing Transformer’s parallel model to address the challenge of extracting multimodal feature information. End-to-end combinatorial deep learning methods overcome limitations present in decision-level analytic models. They explore the correlations among multimodal data, delve deeper into latent features, and enhance the performance of deep learning models, thereby becoming the mainstream approach today. However, despite their advantages, end-to-end combinatorial methods still possess certain shortcomings. For example, feature extraction models such as CNN and RNN employed in end-to-end combinatorial methods excel at handling high-dimensional data swiftly. Nonetheless, they exhibit weaker capability in extracting features from low-resource modal data, leading to subpar model recognition rates (Wang et al., 2022).

Moreover, the dissimilarity in feature scales across non-aligned modal data poses challenges in the alignment fusion process, potentially resulting in the loss of critical feature information. Furthermore, long-term dependency mechanism models like Transformer boast computational efficiency and facilitate single-modal long-term dependency. However, they cannot effectively handle multimodal data and establish multimodal long-term dependency mechanisms (Cukurova, Giannakos & Martinez-Maldonado, 2020). Resolving these challenges and devising more accurate multimodal sentiment analysis models constitute the focal point of our research.

This research article presents an innovative multimodal sentiment analysis model that incorporates a cross-modal Transformer and a lightweight attention aggregation module. The specific contributions of this study are delineated as follows: The utilization of GRU in place of the original Bi-GRU, along with the introduction of a modified algorithm within the lightweight attention aggregation module, enhances the extraction of sentiment features from modal sentiment data. This addresses the limitations of existing models in accurately identifying sentiment features within Dizi meta-modal data.

The proposed convolutional position coding technique effectively reshapes multimodal feature data, facilitating the fusion of multimodal pairs that were previously impeded by noise variations across different data scales.

By enhancing the multiheaded attention mechanism of the conventional Transformer and incorporating a cross-modal multiheaded attention mechanism, the model surmounts the inherent limitations of the traditional Transformer when handling multimodal tasks.

Related work

Previous research in unimodal analysis has predominantly concentrated on investigating the relationship between less widely spoken languages and emotions, studying their associations with more commonly spoken languages like Chinese and English. In contrast to textual data, speech and video data usually require feature engineering methods for extraction. These methods encompass techniques such as mel-frequency cepstrum coefficient (MFCC), wavelet transform, and perceptual linear prediction (PLP), which have been pivotal in feature extraction from modal data.

However, as the demand for recognizing multimodal data features has grown, various approaches have emerged to tackle the challenges of extracting features from low-resource modal data. For instance, Yu et al. (2021) introduced a multimodal data feature extraction model based on BERT and sLSTM. In this model, BERT is employed to extract features from textual modal data, while separate sLSTM models are used for speech and video modal data. A linear model is subsequently applied to comprehensively extract features, resulting in low-resource modal feature data with a more comprehensive representation of essential feature information. Gul et al. (2017) introduced a multimodal adaptation gate that enables BERT and XLNet to extract and dominate low-resource modal data during fine-tuning, adapting textual modal data to mapping changes. Furthermore, Guo et al. (2022) proposed an ER-MRL model for sentiment analysis based on multimodal representation learning. This model employs a neural network-based encoder to vectorize multimodal data, utilizes a gate mechanism for multimodal feature selection, and fully incorporates a Transformer architecture to exploit features from low-resource modal data.

While these models exhibit advantages in extracting features from low-resource modal data, they also present challenges such as complex model structures, many parameters, and improved training efficiency.

Multimodal data, including text, speech, and images, are distinct in measuring emotion features. Text data emphasizes word meanings’ positive and negative polarity; speech data focuses on pitch and intonation patterns, and video data centers around recognizing facial expressions (Baevski et al., 2020). Traditional approaches for aligning and fusing multimodal data often suffer from data loss. Wang et al. (2022) introduced a temporal graph convolutional network (TGCN) for multimodal emotion recognition to address this challenge. TGCN constructs modality-specific graphs and assigns weights to edges based on the distance between multimodal data features, enabling the learning and utilization of embedded information with sequential semantics.

Similarly, Hazarika, Zimmermann & Poria (2020) proposed a framework called MISA, which maps each modality to two different subspaces. These studies employ complex architectures or networks to exploit the alignment information between multimodal data and achieve comprehensive data fusion. However, they encounter performance issues related to the scale and number of parameters (Nezami et al., 2022). Temporal coding techniques, as utilized in traditional convolutional neural networks (Li et al., 2021) and Transformer models (Han et al., 2021), facilitate effective alignment of multimodal data and introduce temporal information into the overall data representation. This incorporation of time series information can enhance training efficiency and model performance. We aim to explore how to combine these temporal coding techniques, building upon their knowledge, to improve the feature extraction capabilities and enhance the accuracy of sentiment analysis models.

A multimodal student sentiment analysis model based on vit

Multimodal data captures sentiment using distinct characteristics across speech, image, and text. Features like pitch and speech rate characterize speech data, while text data focuses on word meanings’ positive and negative polarity. On the other hand, image data is characterized by the recognition of human expressions. Moreover, each modality represents these features differently. Text representations typically involve vectors in a vector space, with dimensions defined by text length and word vector length.

On the other hand, image representations are two-dimensional and represented by pixel values and the semantic information between pixels. Audio representations are defined in terms of period and word vector dimensions. These differences contribute to the non-alignment of multimodal data.

This article proposes a multimodal student sentiment analysis model based on the Visual Transformer (ViT) architecture. The overall model architecture is depicted in Fig. 1.

Figure 1 The architecture of student emotion analysis model.

The first module is the Sentiment Feature Extraction module. It aims to receive low-resource modal data and extract its feature information, producing a tensor as output. This module also incorporates attention weights to enhance the feature information, addressing the model’s limited capability to extract features from low-resource modal data.

The second module is the cross-modal Transformer module. It consists of blocks that align and fuse multimodal information while establishing long-term dependencies between multiple modalities. This module enables the integration and interaction of different modalities, facilitating a comprehensive understanding of the sentiment expressed.

The third module is responsible for sentiment polarity output. It takes as input the tensor outputs from the low-resource modal feature extraction module and the cross-modal Transformer module. These outputs are combined and processed through a fully connected layer, resulting in a floating-point number. The nearest label (an integer) to this floating-point number is selected as the sentiment category label for model prediction.

These modules work together to extract and fuse multimodal features, establish long-term dependencies, and generate sentiment predictions based on the input data.

Sentiment feature extraction module

In multimodal sentiment analysis methods, data alignment plays a crucial role in scaling and aligning modal features. However, this alignment process often results in the loss of vital feature information. To address this issue, we introduce the lightweight attention aggregation module, primarily comprising a bi-directional gated recurrent unit (Bi-GRU) neural network. This module takes in speech and video data, capturing feature information and offering potential alignment cues for subsequent data processing. Its key advantage lies in its efficient extraction of information from low-resource modal data, eliminating the need for extensive data alignment.

Furthermore, the lightweight attention aggregation module contributes low-resource modal data features to the overall model, allowing for fine-tuning the accuracy of the final output. Consequently, inputting low-resource modal data into this module does not necessitate overly complex processing, resulting in enhanced efficiency in feature extraction. Figure 2 provides an illustration of the modal sentiment feature extraction module’s structure.

Figure 2 Structure diagram of modal feature extraction module.

The primary objective of this module is to extract feature information from resource-limited modal data. In the figure, the input data consists of unprocessed images and raw speech data. The dashed data within the module signifies parameter updates, while the resultant output data comprises the respective 1D feature tensors for speech and images. The module encompasses a data processing layer and a feature generation layer. The data processing layer includes a convolutional layer, layer normalization, and a data mapping section, while the remaining data constitutes the feature generation layer.

The feature processing layer prepares the data for feature extraction in the lower layer. The following calculation process uses image data I and speech data A as examples. The data A passes through a convolutional layer with parameters from the Wav2Vec model. In contrast, the data I passes through a convolutional layer with only one parameter randomly generated because audio data needs to be transformed into waveform features before the model can process it. The convolutional squared data method yields the tensor ZA, with the operator expression shown in Eq. (1), where TA represents the total number of speech samples and dA is the tensor dimension of a single speech sample. (1) ZA=CNNθ(A)={Z1,Z2,...,ZTA;Z∈RdA}

The data normalization layer offers the benefit of enhancing the convergence speed of the model while remaining unaffected by the sample size, thereby rendering it suitable for a broader array of scenarios. Data mapping, accomplished through linear regression, is an efficient approach for preprocessing data by partitioning distinct types using a hyperplane. This enables rapid model convergence. The calculation process is illustrated in Eq. (2). (2) PA=Projection(LA)=WΦT×LA+bΦ

where WΦ is used to determine the direction of the hyperplane, the bias bΦ represents the offset of the hyperplane and PA is the refined feature data after data mapping.

The feature generation layer leverages the effectiveness of the GRU model to extract features efficiently. It employs the attentional feature enhancement method and the one-dimensional information aggregation technique to extract crucial feature information from low-resource modal data. With the attention weights determined, the model computes a weighted sum of the features extracted by the GRU. This means that the features that are deemed more important (according to the attention weights) contribute more to the final representation, while less important features have a smaller influence. Generating a one-dimensional information aggregation matrix enhances the significant information of the low-resource modal data, thereby mitigating the risk of information loss.

The feature generation layer receives the mapped data and performs several rounds of operations with the GRU to generate the coarse feature information matrix MΦ, as shown in Eq. (3) and the vector representation of MΦ are shown in Eq. (4), (3) MΦ=GRUθGRU−A(PA)

(4) MΦ={ϕ1,ϕ2,...,ϕdA},i∈[1,dA]

where is the random initialization parameter of the GRU model and is the vector corresponding to each feature information in the coarse feature information matrix.

Cross-modal transformer module

The proposed cross-modal Transformer employs a two-by-two multimodal fusion approach. It utilizes text data as the target matrix and maps speech and video to text, resulting in two cross-modal mapping tensors. These mapping tensors are fused using the same approach to generate a tri-modal mapping tensor. In the multilayer training structure, the cross-modal Transformers employ a feed-forward fusion process to merge the multimodal sequences. Each modality (e.g., images and text) is usually processed separately, and the data from each modality is represented as tensors. These tensors could represent features extracted from the data. Once the mapping is learned, the tensors from different modalities can be combined or fused in various ways. The specific fusion method depends on the problem at hand. For example, you might concatenate the tensors, perform element-wise operations, or use attention mechanisms to give different modalities different weights.

After fusion, the combined tensor is typically passed through additional layers of a neural network or other processing steps to make predictions, classify objects, or perform some other task.Each cross-modal Transformer learns the attention between two modal features while leveraging the underlying features of another modality to enhance the target information repeatedly. This enables effective multimodal fusion and efficient utilization of multimodal feature information. Figure 3 illustrates the structure of the cross-modal Transformer module, with the input data comprising text, speech, and image data, respectively. The output data consists of the cross-modal mapping tensor, and the arrows in the figure indicate the data transfer.

Figure 3 Structural of cross-modal transformer module.

The module commences with a data preprocessing layer, employing a convolutional reshape data technique and additional positional coding. Convolutional reshaping involves utilizing text modalities as the target modal matrix and low-resource modalities as the mapping modal matrix. It also establishes a fusion relationship by leveraging a multiheaded attention mechanism to capture feature information. The additional position encoding method incorporates position encoding to incorporate information regarding the relative or absolute position of each element within the sequence. This is represented as demonstrated in Eqs. (5) and (6). (5) PE(pos,2i)=sin⁡(pos/10,0002i/dmodel)

(6) PE(pos,2i)=cos⁡(pos/10,0002i/dmodeel)

where pos is the position of a token in the sequence, i represents the currently used formula selected by the parity of pos, and dmodeel represents the embedding dimension.

Subsequently, a cross-modal multiheaded attention layer is employed to facilitate the establishment of long-range dependencies. Each element is projected into a distinct space through the dot product of Query, Key, and Value, enabling the extraction of diverse feature information. The multilayer training method is an optimization technique for the model, enabling the fitting of complex functions while effectively reducing the number of network parameters and significantly improving model learning efficiency. Feed-forward neural networks, with their nonlinear learning capabilities and the ability to discover nonlinear features throughout the network, facilitate the efficient extraction of multiheaded information and optimization of feature information.

Since q and k are independently distributed, their expectations are both 0 and their variances are both 1, as shown in Eqs. (7) to (9), where the value of dk is the total number of elements. (7) E(Z)=E(Xq⋅Xk)=E(Xq)E(Xk)=0

(8) D(Z)=D(Xq⋅Xk)=D(Xq)D(Xk)=1

(9) D(∑i=1dkZi)=∑i=1dkD(Zi)=dk

where E(Z) is the expectation (mean) of the random variable Z, and D(Z) is the variance of the random variable Z.

Emotional level output module

The effective polarity output module incorporates an additional cross-modal Transformer module, a Concat layer, a convolutional layer, a pooling layer, a fully connected layer, and an output layer. The structure of this module is illustrated in Fig. 4. Once the output from the cross-modal Transformer is received, the data proceeds to the emotional polarity output module. This module encompasses a Concat layer, a convolutional layer, a pooling layer, a fully-connected layer, and an output layer.

Figure 4 Structure of emotional level output module.

The data after the cross-modal Transformer is the information superimposed on the three modes CMA−>I,I−>L(L,A,V), which is superimposed on the trimodal information. In the Concat layer, CA and CI are tensor stitched with CMA−>I,I−>L(L,A,V), convolved, pooled and then output as a floating point number of samples in the fully connected layer.

The final output of the model is a floating point number between [−3, 3]. The seven integers in the interval represent the sentiment polarity predicted by the model, which also allows the sentiment level of the model output to be judged by its positivity or negativity, meaning that the method in this article can perform both dichotomous and multiclassification tasks. The model output data will be compared with the data labels to evaluate the model performance. Feature fusion methods are an essential approach in the field of pattern recognition. Feature fusion methods can combine the features of multiple modalities to achieve the advantages of various features to complement each other. They can obtain recognition results with more robustness and accuracy.

Experimentation and analysis

As proposed in this manuscript, a range of experiments were conducted to validate the efficacy of the multimodal sentiment analysis model. The experimental setup is detailed in Table 1, while the model’s hyperparameters utilized throughout the experiments are presented in Table 2. It is worth noting that Dropout_1 and Dropout_2 in Table 2 correspond to the Dropout ratios employed in the feed-forward neural network and the fully connected layer, respectively. Additionally, Lr represents the Learning Rate, which is dynamically adjusted in this study, thus expressed as an interval range.

Table 1 Experimental environment.

Environment	Parameter	
OS	Ubuntu 16.04	
CPU	Intel XEON E5-2603	
GPU	NVIDIA Tesla V100	
RAM	32 GB	
Development language	Python 3.8	
Framework	Pytorch 2.0	

Table 2 Hyperparameter setting.

Hyperparameter	Value	
Dropout_1	0.5	
Dropout_2	0.5	
Lr	(1e−2, 1e−4)	
Epochs	500	
Batch_size	128	
Momentum	0.9	

The dataset is from the views of college students on online teaching in colleges and universities during the COVID-19 period (DOI 10.5281/zenodo.4415784). A multimodal approach involving text, speech, and image modalities is used to capture student expressions. Specifically, student reactions within the classroom environment contribute to creating the image dataset, while audio recordings of students responding to questions and providing feedback on the teacher’s instruction are collected for the speech data. Furthermore, students’ reflections on the lesson are gathered to form the text data. Table 3 presents a comprehensive overview of the organization and scale of these three distinct data types. Consequently, the data are categorized into five emotional states: happiness, anger, sadness, surprise, and fear.

Table 3 Organization of dataset.

Data type	People number	Data size	
Text	2,378	2,341,342 (character)	
Image	4,676	21,876	
Audio	4,329	1,876 (min)	

Evaluation indicators

The confusion matrix, referred to as the likelihood matrix or the error matrix, is a valuable visualization tool commonly employed in supervised learning. It encompasses four distinct states: true positive (TP) denotes positive samples correctly predicted as positive, false negative (FN) represents positive samples mistakenly predicted as negative, false positive (FP) signifies negative samples erroneously predicted as positive, and true negative (TN) corresponds to negative samples correctly predicted as negative. From these states, metrics such as recall and accuracy can be derived. Furthermore, the evaluation index of choice, Acc, assesses the overall proportion of accurate classifications. The F1 score captures the comprehensive performance of multi-classification tasks, while MAE quantifies the proportion of misidentifications. Lastly, Corr gauges the level of correlation between two components of a two-dimensional random variable. Corresponding formulas are as follows:

Acc = (Number of Correct Predictions)/(Total Number of Predictions).

Precision = (TP)/(TP + FP).

Recall = (TP)/(TP + FN).

F1 Score = 2 * (Precision * Recall)/(Precision + Recall).

MAE = Σ |Actual − Predicted|/n (summing over all data points).

Coor = Σ [(X − X¯) * (Y − Y¯)]/[√Σ(X − X¯)² * Σ(Y − Y¯)²].

where X and Y are the variables of interest, X¯ and Y¯ are their respective means, and Σ denotes summation over all data points.

Ablation experiments

To evaluate the efficacy of the proposed modal feature extraction module (MFEM), an ablation experiment is conducted, denoted as MFEM hereafter. In this experiment, common feature extraction modules such as CNN, RNN, and LSTM are selected as replacements for MFEM. Moreover, to assess the effectiveness of the proposed Trans-modal Transformer module, the following modules are chosen for ablation experiments: RAVEN, BBFN, and Tri-TransModality. Additionally, for comparison, the Attention Gating structure (AG), LSTM memory module (MM), and Attention Fusion Module (AFM) are selected. These modules aim to verify the effectiveness of the cross-modal multiheaded attention module. The outcomes of the experiments are presented in Tables 4– 6.

Table 4 Ablation experiment results for MFEM.

Modules	Recall	Precision	Acc	F1	MAE	Corr	
CNN	0.7821	0.7941	0.8210	0.8101	0.0145	0.0045	
RNN	0.7954	0.8071	0.8312	0.8156	0.0123	0.0048	
LSTM	0.8321	0.8132	0.8576	0.8394	0.0098	0.0056	
Ours	0.8587	0.8365	0.8890	0.8754	0.0087	0.0070	

Table 5 Ablation experiment results for cross-modal transformer module.

Modules	Recall	Precision	Acc	F1	MAE	Corr	
RAVEN	0.7385	0.7721	0.7952	0.8021	0.0152	0.0051	
BBFN	0.7421	0.7889	0.8021	0.8241	0.0133	0.0053	
Tri-transmodality	0.8143	0.8021	0.8513	0.8301	0.0102	0.0055	
Ours	0.8587	0.8365	0.8890	0.8754	0.0087	0.0070	

Table 6 Ablation experiment results for verifying cross-modal multi-head attention mechanism.

Modules	Recall	Precision	Acc	F1	MAE	Corr	
AG	0.8101	0.8012	0.8274	0.8198	0.0131	0.0054	
MM	0.8231	0.8114	0.8378	0.8331	0.0116	0.0058	
AFM	0.8376	0.8201	0.8612	0.8598	0.0107	0.0062	
Ours	0.8587	0.8365	0.8890	0.8754	0.0087	0.0070	

Based on the data presented in the table, it is evident that all three modules proposed in this article outperform their counterparts across six evaluation metrics. This superiority can be attributed to the design of the trans-modal feature extraction, which exhibits enhanced feature fusion capabilities compared to traditional methods. Notably, the performance of the proposed cross-modal Transformer surpasses that of the second-best performing module, Tri-TransModality, by margins of 0.0444, 0.0344, 0.0377, 0.0453, 0.0015, and 0.0015 across the evaluation metrics. This significant difference underscores the effectiveness of the cross-modal fusion design in sentiment recognition. Furthermore, the proposed cross-modal multiheaded attention module outperforms the second-best performing AFM module by margins of 0.0211, 0.0164, 0.0278, 0.0156, 0.0020, and 0.0008 across the six metrics. This outcome suggests that our attention mechanism approach, incorporating the cross-modal structure, enables the model to focus on the most salient aspects of each modality better.

Comparative experiments

Several comparable approaches were selected for evaluation to ascertain the effectiveness of the proposed cross-modal student sentiment analysis model. These approaches are as follows:

CTC + RAVEN: This approach utilizes parallel LSTM models to process multimodal data independently. It incorporates an attention-gating module and employs the word vector alignment technique to align the multimodal data.

BBFN: This approach employs LSTM memory and cyclic self-connection modules to model the multimodal data. Gating modules are utilized to enhance model training.

CubeMLP: This approach employs separate LSTM models to process sound and image data. The features of the text modal data are abstracted using a transformer encoder. Finally, an MLP fusion network is employed to fuse the multimodal information.

CM-BERT: This approach fine-tunes the BERT model by leveraging the interaction between text and audio. A multimodal attention mechanism dynamically adjusts words’ weight using text and audio information.

The results of the comparative experiments are presented in Table 7.

Table 7 Comparative results.

Methods	Recall	Precision	Acc	F1	MAE	Corr	
CTC+RAVEN	0.7821	0.7654	0.8576	0.8301	0.0114	0.0056	
BBFN	0.7965	0.7712	0.8677	0.8398	0.0112	0.0059	
CubeMLP	0.8097	0.8076	0.8679	0.8421	0.0098	0.0062	
CM-BERT	0.8321	0.8165	0.8721	0.8576	0.0093	0.0063	
Ours	0.8587	0.8365	0.8890	0.8754	0.0087	0.0070	

The tabular data provided in this study unequivocally demonstrates the consistent superiority of the method expounded within this research article when juxtaposed with analogous approaches, as elucidated by a comprehensive evaluation involving six distinct metrics. These empirical findings underscore the remarkable efficacy of the proposed method in the realm of information extraction from multimodal data, coupled with the precision it offers in the domain of sentiment analysis, through the strategic amalgamation of multimodal techniques and Transformer-based architectures. The intrinsic aptitude of the method to proficiently dissect and analyze information stemming from diverse modalities is undeniably manifest.

Furthermore, the elucidation of these findings is supplemented by Fig. 5, which illustrates the confusion matrix stemming from the application of the proposed method to a self-constructed dataset. Notably, this visual representation unveils a sparse distribution of misclassified instances, scattered sporadically across the spectrum of five distinct sentiment categories. Significantly, the preponderance of these misclassifications gravitates toward the vicinity of the diagonal axis, suggestive of the method’s resilience and robust generalization capacity. It is worth emphasizing that such misclassifications are few and specific, indicating that the proposed method’s susceptibility to erroneous categorization is confined to isolated and atypical instances within the dataset.

Figure 5 Confusion matrix.

Conclusion

This article introduces an innovative approach designed to tackle the intricate task of analyzing student emotions within the context of online classroom teaching. Emotion analysis within this setting is crucial for understanding and improving the online learning experience. The proposed method harnesses the power of a visual Transformer and the fusion of multimodal features to achieve accurate and efficient emotion analysis. To accomplish this, the article introduces an emotion modality feature extractor specifically crafted to extract multimodal information effectively and precisely. Additionally, a cross-modal Transformer, which extends upon the Transformer architecture, is presented. This Transformer facilitates the fusion of multimodal features through a two-by-two fusion approach, enhancing the model’s ability to understand the complex interplay of various emotions in the online classroom environment. Ablation experiments are meticulously conducted to validate the efficacy of the feature extractor, cross-modal Transformer, and the cross-modal multiheaded attention structure. Furthermore, comprehensive comparisons with existing methods are performed, and the experimental results unequivocally affirm the superiority of the proposed approach, highlighting its practicality and suitability for deployment in online classrooms.

In the realm of future research, the authors outline their intentions to expand the dataset by incorporating a more extensive range of data encompassing student-teacher interactions in online classrooms. Moreover, the article emphasizes the development of additional modalities for sentiment analysis among students, with the ultimate goal of fortifying the model’s robustness and generalization capabilities. These planned enhancements signify a commitment to further advancing the state of emotion analysis in online education.

Supplemental Information

Supplemental Information 1 This is the code.

Click here for additional data file.

Supplemental Information 2 This is the dataset.

Click here for additional data file.

Additional Information and Declarations

Competing Interests

Author Contributions

Data Availability

The author declares that they have no competing interests.

Lan Ding conceived and designed the experiments, performed the experiments, analyzed the data, performed the computation work, prepared figures and/or tables, authored or reviewed drafts of the article, and approved the final draft.

The following information was supplied regarding data availability:

The dataset is available at Zenodo: Ogundokun, R. O., Muhammad, D., Misra, S., & Awotunde, J. B. (2021). Students' Perspective on Online teaching in higher institutions during COVID 19 [Data set]. Zenodo. https://doi.org/10.5281/zenodo.4415784.

The code is available in the Supplemental File.

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
