# Peer review of "Online teaching emotion analysis based on GRU and nonlinear transformer algorithm"

_PeerJ Computer Science, doi:10.7717/peerj-cs.1696_

## Round 0.1 · original submission · Minor Revisions

Dear authors,

Your article has a few remaining issues. We encourage you to address the concerns and criticisms of the reviewer and resubmit your article once you have updated it accordingly.

Best wishes,

Reviewer 1 ·

Basic reporting

Nonlinear models of neural networks demonstrate the ability to extract important attributes from a given target, thus facilitating automatic analysis of classroom emotions. This paper introduces an online aid tool for analyzing emotional states in virtual classrooms, using the concept of multi-modal fusion as a basic element of emotion analysis, which is ultimately of great significance for evaluating students' participation in educational courses. Some content needs to be modified to improve the content of the paper which is explained below and in the sections that follow.

- Formulas (7) to (9) in the third section lack relevant introduction.
- The references should cite professional articles from outstanding journals in the last three years.

Experimental design

- The lack of specific numerical values in the experimental results is not enough to prove the validity of the experiment.
- Author contributions in the introduction section need to follow the same sentence pattern.
- How is the attentional feature enhancement method used in Section 3.1 to extract key feature information realized?
- In a dataset created by the text itself, the text data collects students' reflections on the course, and the authors need to explain the role of such data in the research model.

Validity of the findings

- For the model evaluation indicators in Section 4.1, corresponding formulas should be listed to make them more perfect.
- The author needs to explain the comparison model mentioned in section 4.3 in more detail.

Cite this review as

Reviewer 2 ·

Basic reporting

In order to study automatic emotion analysis in virtual classrooms and classrooms, this paper uses the concept of multi-mode fusion. Students' auditory input, facial expression and text data are used as basic elements for emotion analysis at the same time. Inspired by Transformer algorithm, a cross-modal Transformer algorithm is proposed. In order to enhance the processing of multi-modal information, the experiment verifies the excellent accuracy of this method, but this paper still has the following shortcomings:
A. There are too few keywords to summarize the content of the abstract;
B. The description of the related work is not logical enough, such as 120 lines of cohesion is not enough, I would suggest the author to adjust it;
C. The interpretation of the formula in 3.1 states that the author needs to check in detail for omissions;
D. What role does the cross-modal mapping tensor generated in Section 3.2 play in the subsequent fusion?
E. Some models in this paper do not give full abbreviations, such as CNN, LSTM, etc., the author needs to check and modify;
F. How can the hyperparameters of the model used in the experiment be dynamically adjusted in the range of le-2 and le-4?
G. The introduction of the pre-processing process of the collected data is lacking in the experiment, and the author can add it appropriately;
H. Figure 5 in the experiment only has the confusion matrix of the method in this paper, I would suggest the author to add the confusion matrix of other models for comparison;
I. The conclusion is too short and insufficient to summarize the content of the whole paper.

Experimental design

.

Validity of the findings

.

Cite this review as

---

## Round 0.2 · accepted · Accept

Dear authors,

Thank you for the revision. It appears that all of the reviewers' comments have been clearly addressed. Your article is accepted for publication after the last revision.

Best wishes,

Reviewer 1 ·

Basic reporting

All suggested changes are incorporated

Experimental design

All suggested changes are incorporated

Validity of the findings

All suggested changes are incorporated

Cite this review as

Reviewer 2 ·

Basic reporting

Author has done required changes

Experimental design

no comments

Validity of the findings

no comments

Additional comments

no comments

Cite this review as